DATA RELEASE

# Chromosome-level genome assembly of the common chiton, *Liolophura japonica* (Lischke, 1873)

Hong Kong Biodiversity Genomics Consortium*,†

## ABSTRACT

Chitons (Polyplacophora) are marine molluscs that can be found worldwide from cold waters to the tropics, and play important ecological roles in the environment. However, only two chiton genomes have been sequenced to date. The chiton *Liolophura japonica* (Lischke, 1873) is one of the most abundant polyplacophorans found throughout East Asia. Our PacBio HiFi reads and Omni-C sequencing data resulted in a high-quality near chromosome-level genome assembly of ~609 Mb with a scaffold N50 length of 37.34 Mb (96.1% BUSCO). A total of 28,233 genes were predicted, including 28,010 protein-coding ones. The repeat content (27.89%) was similar to that of other Chitonidae species and approximately three times lower than that of the Hanleyidae chiton genome. The genomic resources provided by this work will help to expand our understanding of the evolution of molluscs and the ecological adaptation of chitons.

**Subjects** Genetics and Genomics, Animal Genetics, Marine Biology

**Submitted:** 11 January 2024

\* Correspondence on behalf of the consortium. E-mail: jeromehui@cuhk.edu.hk

† Collaborative Authors: Entomological experts who validated the dataset and their affiliations appears at the end of the document

Preprint submitted at https://doi.org/10.1101/2024.01.15.575488

Included in the series: *Hong Kong Biodiversity Genomics* (https://doi.org/10.46471/GIGABYTE_SERIES_0006)

## INTRODUCTION

Mollusca is the second largest animal phylum after Arthropoda and is divided into two subphyla, including the Conchifera (Monoplacophora, Bivalvia, Gastropoda, Scaphopoda, and Cephalopoda) and the Aculifera (Polyplacophora, Caudofoveata, and Solenogastres) [1–3]. Within the latter, chitons (Polyplacophora) are thought to be a relatively early diverging group of living molluscs [4]. They play a crucial role in shaping marine communities in both intertidal and subtidal systems worldwide [5, 6]. These "living fossils" are characterised by a highly evolutionary-conserved and unique type of shell formed by eight articulating aragonite plates that protect from environmental threats [7–9]. This biomineralized armour incorporates an unpigmented sensory system known as aesthetes, which is found in all chiton species [10] as a light-sensing adaptation [11]. Some members of the families Schizochitonidae and Chitonidae developed shell eyes with aragonite-based lenses, allowing the light to focus onto the pigmented photoreceptive retina [10, 12]. However, contrary to other molluscs, our understanding of the Polyplacophora is constrained to only two available genomes (*Acanthopleura granulata* and *Hanleya hanleyi*) [13, 14].

*Liolophura japonica* (Polyplacophora, Chitonidae) (Lischke, 1873) (Figure 1A) is one of the most abundant polyplacophorans found on the intertidal rocky reefs of the Asian continent, including China, Korea, and Japan. This species diverged from the last common ancestor of *Liolophura* ~184 million years ago during the early Pleistocene period [15]. On the shore, they are distributed over a wide vertical range from the mid-littoral zone to the

| | *Liolophura japonica* |
|---|---|
| Accesion number | GCA_032854445 |
| total_length | 609,495,693 |
| number | 636 |
| mean_length | 964,392 |
| N_count | 0.0364% |
| Gaps | 1,110 |
| N50 | 37,343,639 |
| N50n | 5 |
| BUSCOs | 96.10% |
| HiFi (X) | 14 |
| HiFi Reads | 1,050,568 |
| HiFi Bases | 8,769,373,110 |
| HiFi Ave_len | 8,347 |
| Gene models | 28,233 |

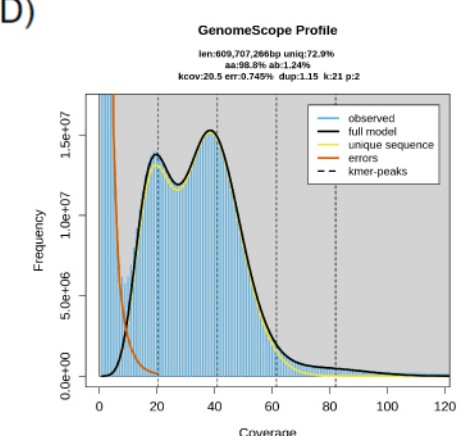

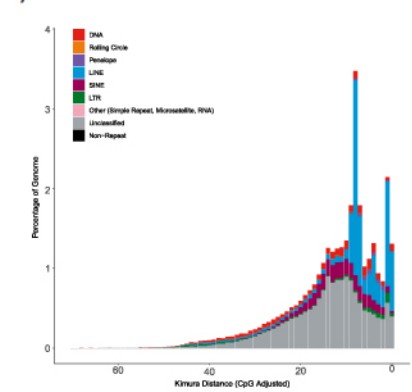

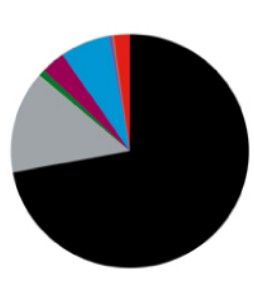

**Figure 1.** (A) Picture of *Liolophura japonica*; (B) Statistics of the genome assembly generated in this study; (C) Hi-C contact map of the assembly visualised using Juicebox (v1.11.08); (D) Genomescope report with k-mer = 21; (E) Repetitive elements distribution.

low-subtidal zone, where the animals experience periodic fluctuations in environmental conditions with the ebb and flow of the tide [16, 17]. Unlike many other mobile species inhabiting rocky shores, which migrate towards the low shore areas during summer [18, 19], *L. japonica* does not show any significant seasonal migration behaviour and can survive stressful low tide periods presumably by fitting itself into small refuges via its eight flexible, interlocking plates [17, 20, 21]. In terms of its feeding biology, it is a generalist consumer feeding on a wide range of microalgae and macroalgae [16, 22] and, owing to their high density, it is an important grazer in the intertidal zone of East Asia, contributing to the control of on-shore primary productivity [22, 23]. Given the importance of this species, the high-quality genome presented in this study will help to expand our understanding of the evolution of molluscs and the ecological adaptation of chitons.

## CONTEXT

Here, we report the assembled genome of the chiton *L. japonica* (Polyplacophora, Chitonidae) (Lischke 1873) (Figure 1A). *L. japonica* was selected to be one of the species sequenced by the Hong Kong Biodiversity Genomics Consortium (also known as

EarthBioGenome Project Hong Kong), which is organised by researchers from eight publicly funded universities in Hong Kong. The *L. japonica* genome presented in this study is of high quality and near chromosomal level, providing a valuable resource for the understanding of the evolutionary biology of polyplacophorans and the adaptation of its resilience under oscillating environmental changes in the intertidal zones.

## METHODS

## Collection and storage of samples, isolation of high molecular weight genomic DNA, quantification, and qualification

The chitons *L. japonica* were collected at the rocky shore in Kau Sai Chau, Hong Kong (22.380 °N, 114.310 °E) during the summer of 2022. They were kept in 35 ppt artificial seawater at room temperature until DNA isolation. High molecular weight (HMW) genomic DNA was isolated from a single individual. The foot muscle was first frozen in liquid nitrogen and ground to powder. DNA extraction was carried out with the sample powder using a Qiagen MagAttract HMW kit (Qiagen Cat. No. 67563) following the manufacturer's protocol with some modifications.

Around 1 g of sample was first put in 200 μl 1× PBS and mixed with RNase A, Proteinase K, and Buffer AL provided in the kit. The mixture was allowed to sit at room temperature (~22 °C) for 2 h. Next, the mixture was gently flicked every 30 min to allow thorough mixing of samples and digestion solution. The DNA was then isolated from the lysate with the magnetic beads provided in the kit and a magnet rack. Finally, the sample was eluted with 120 μl of elution buffer (PacBio Ref. No. 101-633-500). To prevent unintended DNA shearing during the extraction process, wide-bore tips were consistently employed whenever DNA transfer occurred. Next, the sample was quantified by the Qubit® Fluorometer, Qubit™ dsDNA HS, and BR Assay Kits (Invitrogen™ Cat. No. Q32851). Overnight pulse-field gel electrophoresis was used to examine the molecular weight of the isolated DNA, together with three DNA markers (λ-Hind III digest; Takara Cat. No. 3403, DL15,000 DNA Marker; Takara Cat. No. 3582A and CHEF DNA Size Standard-8-48 kb Ladder; Cat. No. 170-3707). The purity of the sample was examined by the NanoDrop™ One/OneC Microvolume UV–Vis Spectrophotometer, with A260/A280: ~1.8 and A260/A230: >2.0 as a standard.

## DNA shearing, library preparation, and sequencing

A total of 120 μl of DNA sample with 6.2 μg DNA was transferred to a g-tube (Covaris Part No. 520079). The sample was then subjected to six centrifugation steps for 2 min each at 2,000 × g. The resultant DNA was collected and stored in a 2 mL DNA LoBind® Tube (Eppendorf Cat. No. 022431048) at 4 °C until library preparation. Overnight pulse-field gel electrophoresis was used to examine the molecular weight of the isolated DNA, as described in the previous section. The electrophoresis profile was set as follows: 5 K as the lower end and 100 K as the higher end for the designated molecular weight; Gradient = 6.0 V/cm; Run time = 15 h:16 min; included angle = 120°; Int. Sw. Tm = 22 s; Fin. Sw. Tm = 0.53 s; Ramping factor: a = Linear. The gel was run in 1.0% PFC agarose in 0.5× TBE buffer at 14 °C. A SMRTbell library was constructed using the SMRTbell® prep kit 3.0 (PacBio Ref. No. 102-141-700), following the manufacturer's protocol.

The genomic DNA was first subjected to DNA repair to remove single-stranded overhangs and repair damages from the shearing step on the DNA backbone. After repair, both ends of

**Table 1.** Details of genome and transcriptome sequencing data.

| Genome sequencing data | | | | |
|---|---|---|---|---|
| **Liabrary** | **No. of reads** | **No. of bases** | **Accession** | **Coverage (X)** |
| PacBio HiFi | 1,050,568 | 8,769,373,110 | SRX20411988 | 14 |
| Omnic | 264,637,334 | 39,695,600,100 | SRX21911526 | 65 |
| **Transcriptome sequencing data** | | | | |
| **Sample name** | **No. of reads** | **No. of bases** | **Accession** | **Tissue type** |
| Lj2HS_Dg_T | 35,899,054 | 5,384,856,396 | SAMN35319765 | Digestive gland |
| Lj2HS_Ft_T | 35,289,414 | 5,293,410,229 | SAMN35319766 | Foot |
| Lj2HS_Gl_T | 34,240,482 | 5,136,070,054 | SAMN35319767 | Gill |
| Lj2HS_Gn_T | 31,128,962 | 4,669,342,254 | SAMN35319768 | Gonad |
| Lj2HS_Ht_T | 37,837,458 | 5,675,616,543 | SAMN35319769 | Heart |

the DNA were polished and tailed with an A-overhang. The ligation of T-overhang SMRTbell adapters was performed at 20 °C for 30 min. Next, the SMRTbell library was purified with SMRTbell® cleanup beads (PacBio Ref. No. 102158-300). The concentration and size of the library were examined using the Qubit® Fluorometer, Qubit™ dsDNA HS, and BR Assay Kits (Invitrogen™ Cat. No. Q32851), and the pulse-field gel electrophoresis, respectively. A subsequent nuclease treatment step was performed to remove any non-SMRTbell structure in the library mixture. A final size-selection step was carried out to remove the small DNA fragments in the library with 35% AMPure PB beads [24]. The Sequel® II binding kit 3.2 (PacBio Ref. No. 102-194-100) was used for the final preparation for sequencing. Sequel II primer 3.2 and Sequel II DNA polymerase 2.2 were annealed and bound to the SMRTbell library, respectively. The library was loaded at an on-plate concentration of 50–90 pM using the diffusion loading mode. The sequencing was conducted on the Sequel IIe System with an internal control provided in the binding kit. The sequencing was set up and performed in 30-hour movies (with 120 min pre-extension) with the software SMRT Link v11.0 (PacBio). HiFi reads were generated and collected for further analysis. One SMRT cell was used for this sequencing. Detailed sequencing data can be found in Table 1.

## Omni-C library preparation and sequencing

An Omni-C library was constructed using the Dovetail® Omni-C® Library Preparation Kit (Dovetail Cat. No. 21005) according to the manufacturer's instructions. Around 20 mg of flash-freezing powered tissue sample was added into 1 mL 1× PBS, where the genomic DNA was crosslinked with formaldehyde, and the fixed DNA was digested with endonuclease DNase I. Next, the concentration and fragment size of the digested sample were checked by the Qubit® Fluorometer, Qubit™ dsDNA HS, and BR Assay Kits (Invitrogen™ Cat. No. Q32851), and the TapeStation D5000 HS ScreenTape, respectively. Following the quality examination, both ends of the DNA were polished. Ligation of a biotinylated bridge adaptor was conducted at 22 °C for 30 min, and the subsequent proximity ligation between crosslinked DNA fragments was performed at 22 °C for 1 h. After the ligation events, the DNA was reverse crosslinked and then purified with SPRIselect™ Beads (Beckman Coulter Product No. B23317) to remove the biotin that was not internal to the ligated fragments. The Dovetail™ Library Module for Illumina (Dovetail Cat. No. 21004) was used for the end repair and adapter ligation. During this process, the DNA was tailed with an A-overhang, which allowed Illumina-compatible adapters to ligate to the DNA fragments at 20 °C for 15 min. The Omni-C library was then sheared into small fragments with USER Enzyme Mix and purified with SPRIselect™ Beads. Next, Streptavidin Beads were added to isolate the DNA

fragments with internal biotin. Universal and Index PCR Primers from the Dovetail™ Primer Set for Illumina (Dovetail Cat. No. 25005) were used to amplify the library. The final size selection step was carried out with SPRIselect™ Beads to pick only the DNA fragments ranging between 350 bp and 1,000 bp. At last, the concentration and fragment size of the sequencing library were examined by the Qubit® Fluorometer, Qubit™ dsDNA HS, and BR Assay Kits, and the TapeStation D5000 HS ScreenTape, respectively. After the quality check, the library was sequenced on an Illumina HiSeq-PE150 platform. Detailed sequencing data information can be found in Table 1.

## Transcriptome sequencing

Total RNA and small RNA (<200 nt) from different tissues (i.e., digestive gland, foot, gill, gonad, and heart) of the other individual were isolated using the TRIzol reagent (Invitrogen) and the mirVana™ miRNA Isolation Kit (Ambion) following the manufacturer's protocol respectively. The quality of the extracted RNA was checked using NanoDrop™ One/OneC Microvolume UV–Vis Spectrophotometer (Thermo Scientific™ Cat. No. ND-ONE-W) and gel electrophoresis. The qualified transcriptome samples were sent to Novogene Co. Ltd (Hong Kong, China) for the library construction for polyA-selected RNA sequencing using the TruSeq RNA Sample Prep Kit v2 (Illumina Cat. No. RS-122-2001), and 150 bp paired-end sequencing. Agilent 2100 Bioanalyser (Agilent DNA 1000 Reagents) was used to measure the insert size and concentration of the final libraries. Details of the sequencing data are listed in Table 1.

## Genome assembly, gene model prediction, repeat analysis, and genome size estimation

*De novo* genome assembly was performed using Hifiasm (version 0.16.1-r375; RRID:SCR_021069) with default parameters [25]. Haplotypic duplications were identified and removed using purge_dups (RRID:SCR_021173) based on the depth of HiFi reads [26]. Proximity ligation data from the Omni-C library were used to scaffold the genome assembly by YaHS (version 1.2a.2) with default parameters [27]. Transposable elements (TEs) were annotated as previously described using the automated Earl Grey TE annotation pipeline (version 1.2) [28]. The mitochondrial genome was assembled using MitoHiFi (v2.2) [29]. RNA sequencing data were first processed with Trimmomatic (version 0.39; RRID:SCR_011848) with parameters "TruSeq3-PE.fa:2:30:10 SLIDINGWINDOW:4:5 LEADING:5 TRAILING:5 MINLEN:25" [30]. Gene models were then trained and predicted by funannotate (version 1.8.15) [31] using the parameters "--repeats2evm --protein_evidence uniprot_sprot.fasta --genemark_mode ET --optimize_augustus --organism other --max_intronlen 350000". Briefly, the genome was soft-masked by redmask (v0.0.2) [32]. In the funannotate-train step, the transcripts assembled by Trinity (v2.8.5; RRID:SCR_013048) with parameters "--stranded RF --max_intronlen 350000" [33] were used to map to the repeat soft-masked genome by minimap2 (v2.2.1; RRID:SCR_018550) with default parameters [34]. Next, the Trinity transcript alignments were converted to the GFF3 format and used as input for the PASA alignment (v2.5.3; RRID:SCR_014656) with default parameters [35] in the Launch_PASA_pipeline.pl process to get the PASA models trained by TransDecoder (v5.7.1; RRID:SCR_017647) with default parameters [36]. Finally, Kallisto TPM data (v 0.46.1; RRID:SCR_016582) [37] was used to select the PASA gene models. The PASA gene models were used to train Augustus (RRID:SCR_008417) in the funannotate-predict step. The gene

models from several prediction sources, including GeneMark (v3.68_lic; RRID:SCR_011930) [38], high-quality Augustus predictions, PASA (v2.5.3) [35], Augustus (v3.5.0) [39], GlimmerHM (v 3.0.4) [40], and SNAP (v2006-07-28; RRID:SCR_007936) [41], were passed to EVidence Modeler (v1.1.1; RRID:SCR_014659) to generate the gene model annotation files. Untranslated regions (UTRs) were captured in the funannotate-update step using PASA. Briefly, PASA was run twice in the funannotate-update step, and the transcripts generated in the previous funannotate-train step were mapped to the genome. These data were automatically parsed and used to update the UTR data using the PASA comparison method according to the PASA built-in process. In total, 10,327 UTRs were updated in this study. The protein-coding genes were searched with BLASTP (RRID:SCR_001010) against the NR and swissprot databases by DIAMOND (v0.9.24; RRID:SCR_016071) [42] with parameters "−−more-sensitive −−evalue $1 \times 10^{-3}$" and mapped by HISAT2 (version 2.1.0; RRID:SCR_015530) [43] with transcriptome reads. A total of 73.9% and 55.8% of the 28,032 protein-coding genes were mapped to the NR and swissprot databases, respectively. The BUSCO (RRID:SCR_015008) of the protein-coding genes was 90.9%, including 83.5% complete and single-copy genes, 7.4% complete and duplicated genes, 5.0% fragmented genes, while 4.1% of the BUSCOs genes were missed (metazoa_odb10 with 954 total BUSCOs genes) (v 5.5.0) [44].

TEs were annotated as previously described [28] using the automated Earl Grey TE annotation pipeline (version 1.2) with "-r eukarya" to search the initial mask of known elements and other default parameters. Briefly, this pipeline first identified known TEs from Dfam (RRID:SCR_021168) with RBRM (release 3.2) and Repbase (v20181026; RRID:SCR_021169). *De novo* TEs were then identified, and the consensus boundaries were extended using an automated BLAST (RRID:SCR_004870), Extract, and Extend process with five iterations and 1,000 flanking bases added at each round. Redundant sequences were removed from the consensus library before the genome assembly was annotated with the combined known and *de novo* TE libraries. Annotations were processed to remove overlap and defragment annotations prior to the final TE quantification.

The 21 k-mer count and histogram were generated using Omni-C reads and Jellyfish (v2.3.0; RRID:SCR_005491) with the parameters "count -C -m 21 -s 1000000000 -t 10" [45]. Next, the reads.histo were uploaded to GenomeScope (v2.0; RRID:SCR_017014) to estimate the genome heterozygosity, repeat content, and size using default parameters [46].

Hi-C contact maps were generated using the Juicer tools (version 1.22.01; RRID:SCR_017226), following the Omni-C manual [47]. Briefly, Omni-C reads were mapped and aligned by BWA (RRID:SCR_010910) with parameters "mem -5SP -T0". Next, the parsing module of the pairtools pipeline (v1.0.2) [48] was used to find ligation junctions with parameters "−−min-mapq 40 −−walks-policy 5unique −−max-inter-align-gap 30 −−nproc-in 8 −−nproc-out 8". The parsed pairs were then sorted using pairtools sort with default parameters, and PCR duplicate pairs were removed using pairtools dedup with parameters "−−nproc-in 8 −−nproc-out 8 −−mark-dups". The pairs' file was generated using pairtools split with default parameters and used to generate the contact matrix using juicertools and Juicebox (v1.11.08; RRID:SCR_021172) [49].

## RESULTS AND DISCUSSION

A total of 8.77 Gb of HiFi bases of the common chiton *L. japonica* were generated with an average HiFi read length of 8,347 bp with 14X data coverage. After scaffolding with ~397 Gb

**Table 2.** Summary of the genome statistics of *Liolophura japonica*, *Acanthopleura granulate* [13], and *Hanleya hanleyi* [14].

|  | *Liolophura japonica* | *Acanthopleura granulata* | *Hanleya hanleyi* |
|---|---|---|---|
| Total length (bp) | 609,495,693 | 606,536,932 | 2,516,608,230 |
| Number | 632 | 87 | 57,495 |
| Mean length (bp) | 964,392 | 6,971,689 | 43,771 |
| Longest | 96,433,062 | 50,906,754 | 801,547 |
| Shortest | 1,000 | 43,755 | 1,532 |
| N count | 0.0364% | 10.1930% | 0.0005% |
| Gaps | 1,110 | 5,849 | 216 |
| N50 | 37,343,639 | 23,921,462 | 65,037 |
| N50n | 5 | 9 | 10,427 |
| BUSCO (genome, metazoa_odb10) | C:96.1% [S:95.2%, D:0.9%], F:1.9%, M:2.0%, n:954 | C:96.0% [S:95.3%, D:0.7%], F:2.1%, M:1.9%, n:954 | C:80.0% [S:74.9%, D:5.1%], F:12.1%, M:7.9%, n:954 |
| Total length of protein-coding genes (AA) | 12,529,260 | 17,148,789 | 22,920,296 |
| Number of protein-coding genes | 28,010 | 37,872 | 69,284 |
| Mean length of protein-coding genes (AA) | 447 | 453 | 331 |
| BUSCO (Proteome, metazoa_odb10) | C:90.9% [S:83.5%, D:7.4%], F:5.0%, M:4.1%, n:954 | C:93.4% [S:71.4%, D:22.0%], F:2.8%, M:3.8%, n:954 | C:81.8% [S:75.6%, D:6.2%], F:11.1%, M:7.1%, n:954 |
| Reference | This study | [13] | [14] |

**Table 3.** Scaffold information of 13 pseudomolecules.

| Chr_number | scaffold id | scaffold length | Cumulative % of the whole genome |
|---|---|---|---|
| 1 | scaffold_1 | 96,433,062 | 15.82% |
| 2 | scaffold_2 | 82,493,220 | 29.35% |
| 3 | scaffold_3 | 60,359,117 | 39.25% |
| 4 | scaffold_4 | 58,388,612 | 48.83% |
| 5 | scaffold_5 | 37,343,639 | 54.96% |
| 6 | scaffold_6 | 36,638,032 | 60.97% |
| 7 | scaffold_7 | 32,819,725 | 66.35% |
| 8 | scaffold_8 | 31,077,156 | 71.45% |
| 9 | scaffold_9 | 30,102,928 | 76.39% |
| 10 | scaffold_10 | 28,395,274 | 81.05% |
| 11 | scaffold_11 | 27,489,247 | 85.56% |
| 12 | scaffold_12 | 25,057,479 | 89.67% |
| 13 | scaffold_13 | 19,479,325 | 92.86% |

Omni-C data, the assembled genome size was 609.5 Mb, with 632 scaffolds, a scaffold N50 of 37.34 Mb, and the complete BUSCO estimation of 96.1% (metazoa_odb10) (Figure 1B; Table 2). A total of 13 pseudomolecules of chromosomal length were anchored from the Omni-C data (Figure 1C; Table 3). This result is close to the karyotype of *L. japonica* ($2n = 24$), indicating the assembly is near chromosome-level. The assembled *L. japonica* genome has a genome size close to the estimation performed by GenomeScope, which was 609.7 Mb with a heterozygosity rate of 1.24% (Figure 1D; Table 4), and similar to the *Acanthopleura granulata* chiton genome (*A. granulate*: 606 Mb) [13] (Table 2). Telomeres can also be found in 7 out of 13 pseudomolecules (Table 5).

Total RNA sequencing data from different tissues, including the digestive gland, foot, gill, gonad, and heart, was used to assemble the transcriptome of *L. japonica*. The final transcriptome assembly contained 294,118,260 transcripts, with 192,010 Trinity-annotated genes (average length of 1,100 bp and N50 length of 2,373 bp). The resultant transcriptomes

**Table 4.** GenomeScope statistics report at K-mer = 21.

| Property | min | max |
|---|---|---|
| Homozygous (aa) | 98.74% | 98.78% |
| Heterozygous (ab) | 1.22% | 1.26% |
| Genome haploid length (bp) | 608,187,534 | 609,707,265 |
| Genome repeat length (bp) | 165,056,812 | 165,469,254 |
| Genome unique length (bp) | 443,130,722 | 444,238,011 |
| Model fit | 78.27% | 99.09% |
| Read error rate | 0.74% | 0.74% |

**Table 5.** List of telomeric repeats found in nine scaffolds.

| Scaffold id | Strand | Position | Sequence |
|---|---|---|---|
| scaffold_1 | Forward | start | CTAACCCTAACCCTAACCCTCACCTAACCCCTAACCCTAACCCTAACCCT |
| scaffold_2 | Reverse | end | TAGGGTTAGGGTTAGGGTTAGGGTTAGGGTTAGGGTTAGGGTTAGGGTTA |
| scaffold_5 | Forward | start | CTAAACCCTAACCCTAACCCTAACCCTAACCCTAACCCTAACCCTAACCC |
| scaffold_6 | Forward | start | CTAACCCCTAACCCTAACCCCTAACCCTAAACCCAAACCCTAACCTAACC |
| scaffold_6 | Reverse | end | TAGGGTTAGGGTTAGGGTTAGGGTTAGGGTTAGGGTTAGGGTTAGGGTTA |
| scaffold_7 | Forward | start | TATCCCTAACCCTAACCCTATACCCTAAACCCTAACCCTAACCCTATACC |
| scaffold_8 | Forward | start | CCTAACCCTAACCCTAACCCTAACCCTAACCCTAACCCTAACCCTAACCC |
| scaffold_9 | Forward | start | AACCCTAACCCTAACCCTAACCCTAACCCTAACCCTAACCCTAACCCTAA |
| scaffold_9 | Reverse | end | GGGTTTAGGGTTAGGGTTAGGGTTAGGGTAGGGTTAGGGTTAGGGTTAGG |

**Table 6.** Catalogue of repeat elements in the *Liolophura japonica* genome.

| Classification | Total length (bp) | Count | Proportion (%) | No. of distinct classifications |
|---|---|---|---|---|
| DNA | 13,496,516 | 29,294 | 2.21 | 5,097 |
| LINE | 40,857,525 | 12,510 | 6.70 | 4,445 |
| LTR | 5,801,101 | 8,948 | 0.95 | 2,645 |
| Other (simple repeat, microsatellite, RNA) | 471,212 | 641 | 0.08 | 408 |
| Penelope | 3,180,531 | 6,938 | 0.52 | 2,484 |
| Rolling circle | 896,188 | 1,714 | 0.15 | 968 |
| SINE | 19,737,403 | 85,584 | 3.24 | 1,124 |
| Unclassified | 85,585,554 | 221,825 | 14.04 | 6,188 |
| SUM: | 170,026,030 | 367,454 | 27.89 | 23,359 |

were used to predict the gene models, and 28,233 gene models were generated with 28,010 predicted protein-coding genes, a mean coding sequence length of 447 amino acids (AA), and the proteome complete BUSCO estimation of 90.9% (metazoa_odb10) (Table 2).

For the repeat elements, a total repeat content of 27.89% was found in the genome assembly, including 14.04% of unclassified elements. This result is comparable to the estimated genome repeat length with kmer 21 (27.14%) and the Chitonidae species *A. granulata* (23.56%) [13] (Figure 1E; Tables 6, 4). Among the remaining repeats, long interspersed nuclear elements (LINEs) were the most abundant (6.70%), followed by short interspersed nuclear elements (SINEs) (3.24%) and DNA (2.21%), whereas long terminal repeats (LTRs), Penelope, rolling circle, and others were only present in low proportions (LTR: 0.95%, Penelope: 0.52%, rolling circle: 0.15%, other: 0.08%).

## DATA VALIDATION AND QUALITY CONTROL

To assess the quality of samples in DNA extraction and PacBio library preparation, NanoDrop™ One/OneC Microvolume UV–Vis Spectrophotometer, Qubit® Fluorometer, and overnight pulse-field gel electrophoresis were performed. Furthermore, the Omni-C library quality was assessed by Qubit® Fluorometer and TapeStation D5000 HS ScreenTape.

During genome assembly, the Hifiasm output was mapped with PacBio HiFi reads with minimap2 (v2.15-r905) with parameters "-ax asm20" (Li, H. 2018) and compared to the NT database via BLAST with parameters "-task megablast -outfmt '6 qseqid staxids bitscore std' -max_target_seqs 1 -max_hsps 1 -evalue $1 \times 10^{-25}$", which was used as the input for Blobtools (v1.1.1; RRID:SCR_017618) [50]. Scaffolds recognised as possible contaminations were removed from the assembly (Figure 2). The GC content ranged from 27.73% to 54.34%, and the coverage ranged from 0.9576 to 216.3843 for the remaining contigs. Moreover, a kmer-based statistical approach was employed to estimate the genome heterozygosity, repeat content, and genome size from sequencing Omni-C reads using Jellyfish (v2.3.0) [45] and GenomeScope (v2.0) [46] (Figure 1D, Table 4). In addition, telomeric repeats were screened by using FindTelomeres [51]. BUSCO (v5.5.0) [44] was used to assess the completeness of the genome assembly and gene annotation with a metazoan dataset (metazoa_odb10).

## CONCLUSION AND FUTURE PERSPECTIVE

The high-quality, near chromosome-level *L. japonica* genome presented in this study is a valuable resource for gaining new insights into the environmental adaptations of *L. japonica* in residing the intertidal zones and for future investigations on the evolutionary biology in polyplacophorans and other molluscs.

## DATA AVAILABILITY

The final assemblies, Omni-C data, and PacBio HiFi reads were submitted to NCBI under the accession number GCA_032854445.1. The raw reads generated in this study were deposited to the NCBI database under the BioProject accession PRJNA973839, with PacBio HiFi reads SRX20411988, Omni-c reads SRX21911526 and transcriptome data (SAMN35319765, SAMN35319766, SAMN35319767, SAMN35319768, SAMN35319769). The genome annotation files were deposited and are publicly available in figshare [52].

## ABBREVIATIONS

AA, amino acids; HMW, high molecular weight; LINE, long interspersed nuclear element; LTR, long terminal repeat; SINE, short interspersed nuclear element; TE, transposable elements; UTR, untranslated region.

## DECLARATIONS

### Ethics approval and consent to participate

The authors declare that ethical approval was not required for this type of research.

### Competing interests

The authors declare that they have no competing interests.

blobtools_pacbio.blobDB.json.bestsum.phylum.p8.span.100.blobplot.bam0

**Figure 2.** Genome assembly quality control and contaminant/cobiont detection for the *Liolophura japonica.*

## Authors' contributions

JHLH, TFC, LLC, SGC, CCC, JKHF, JDG, SCKL, YHS, CKCW, KYLY, and YW conceived and supervised the study. MFFA and WLS carried out the DNA extraction, library preparation, and sequencing. WN performed the genome assembly and gene model prediction. MFFA conducted the homeobox gene annotation. TYH, BKHL, and GAW collected the chiton samples.

## Funding

This work was funded and supported by the Hong Kong Research Grant Council Collaborative Research Fund C4015-20EF, CUHK Strategic Seed Funding for Collaborative Research Scheme (3133356), and CUHK Group Research Scheme (3110154).

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

## DETAILS OF COLLABORATIVE AUTHORS

### • List of authors in Hong Kong Biodiversity Genomics Consortium

Jerome H. L. Hui,[1] Ting Fung Chan,[2] Leo Lai Chan,[3] Siu Gin Cheung,[4] Chi Chiu Cheang,[5] James Kar-Hei Fang,[6] Juan Diego Gaitan-Espitia,[7] Stanley Chun Kwan Lau,[8] Yik Hei Sung,[9,10] Chris Kong Chu Wong,[11] Kevin Yuk-Lap Yip,[12,13] Yingying Wei,[14] Ming Fung Franco Au,[1] Wai Lok So,[1] Wenyan Nong,[1] Tin Yan Hui,[9,15] Brian Kai Hin Leung,[15] Gray A. Williams[15]

[1]School of Life Sciences, Simon F.S. Li Marine Science Laboratory, State Key Laboratory of Agrobiotechnology, Institute of Environment, Energy and Sustainability, The Chinese University of Hong Kong, Hong Kong, China

[2]School of Life Sciences, State Key Laboratory of Agrobiotechnology, The Chinese University of Hong Kong, Hong Kong SAR, China

[3]State Key Laboratory of Marine Pollution and Department of Biomedical Sciences, City University of Hong Kong, Hong Kong SAR, China

[4]State Key Laboratory of Marine Pollution and Department of Chemistry, City University of Hong Kong, Hong Kong SAR, China

[5]Department of Science and Environmental Studies, The Education University of Hong Kong, Hong Kong SAR, China

[6]State Key Laboratory of Marine Pollution, Research Institute for Future Food, and Department of Food Science and Nutrition, The Hong Kong Polytechnic University, Hong Kong SAR, China

[7]The Swire Institute of Marine Science and School of Biological Sciences, The University of Hong Kong, Hong Kong SAR, China

[8]Department of Ocean Science, The Hong Kong University of Science and Technology, Hong Kong SAR, China

[9]Science Unit, Lingnan University, Hong Kong SAR, China

[10]School of Allied Health Sciences, University of Suffolk, Ipswich, IP4 1QJ, UK

[11]Croucher Institute for Environmental Sciences, and Department of Biology, Hong Kong Baptist University, Hong Kong SAR, China

[12]Department of Computer Science and Engineering, The Chinese University of Hong Kong, Hong Kong SAR, China

[13]Sanford Burnham Prebys Medical Discovery Institute, La Jolla, CA, USA

[14]Department of Statistics, The Chinese University of Hong Kong, Hong Kong SAR, China

[15]The Swire Institute of Marine Science and School of Biological Sciences, The University of Hong Kong, Hong Kong SAR, China

