## [Editor Report]

Editor’s AssessmentThis work is part of a series of papers from the Hong Kong Biodiversity Genomics Consortium sequencing the rich biodiversity of species in Hong Kong. This example assembles the genome of the common chiton, Liolophura japonica (Lischke, 1873). Chitons are marine molluscs that can be found worldwide from cold waters to the tropics that play important ecological roles in the environment, but to date are lacking in genomes with only a few assemblies available. This data was produced using PacBio HiFi reads and Omni-C sequencing data, the resulting genome assembly being around 609 Mb in size. From this 28,010 protein-coding genes were predicted. After review improved the methodological details the quality metrics look near chromosome-level, having a scaffold N50 length of 37.34 Mb and 96.1% BUSCO score. This high-quality genome should hopefully be a valuable resource for gaining new insights into the environmental adaptations of L. japonica in residing the intertidal zones and for future investigations in the evolutionary biology in Polyplacophorans and other molluscs.

---

## [Reviewer Report]

Reviewer name and names of any other individual's who aided in reviewer Jin SunDo you understand and agree to our policy of having open and named reviews, and having your review included with the published papers. (If no, please inform the editor that you cannot review this manuscript.)YesIs the language of sufficient quality?YesPlease add additional comments on language quality to clarify if needed
Are all data available and do they match the descriptions in the paper? YesAdditional Commentsthe assembly and annotations can be found in the FigshareAre the data and metadata consistent with relevant minimum information or reporting standards? See GigaDB checklists for examples <a href="http://gigadb.org/site/guide" target="_blank">http://gigadb.org/site/guide</a>YesAdditional CommentsIs the data acquisition clear, complete and methodologically sound?YesAdditional CommentsIs there sufficient detail in the methods and data-processing steps to allow reproduction?YesAdditional CommentsIs there sufficient data validation and statistical analyses of data quality? YesAdditional CommentsIs the validation suitable for this type of data?YesAdditional CommentsI have examined the HiC interaction map, and I think the scaffolding is high-quality. Is there sufficient information for others to reuse this dataset or integrate it with other data?YesAdditional CommentsAny Additional Overall Comments to the AuthorThe presentation is clear, but i would suggest the authors include the BUSCO score for the gene models. RecommendationMinor Revision

---

## [Reviewer Report]

Upload additional filesDRR-202401-02/form/Comments_to_authors.pdfReviewer name and names of any other individual's who aided in reviewer Priscila M SalloumDo you understand and agree to our policy of having open and named reviews, and having your review included with the published papers. (If no, please inform the editor that you cannot review this manuscript.)YesIs the language of sufficient quality?YesPlease add additional comments on language quality to clarify if needed
The language is appropriate and does not hinder understanding, but some minor proof reading could benefit the manuscript. I left a few suggestions in my comments to the authors.Are all data available and do they match the descriptions in the paper? NoAdditional CommentsThe data made available on NCBI has the 632 scaffolds, but the 13 pseudomolecules are not shown (in GCA_032854445.1, under Chromosomes, it reads “This scaffold-level genome assembly includes 632 scaffolds and no assembled chromosomes”), please clarify where information/data for the 13 pseudomolecules can be found.  The figshare repository has the annotation files, but it lacks a metadata file detailing what each of the annotation files is (the file names are descriptive, but they do not replace a metadata file).  The data availability statement lacks information about the transcriptomes (were these made available?) Supplementary tables are mentioned in the text file but were not made available (at least not for review).Are the data and metadata consistent with relevant minimum information or reporting standards? See GigaDB checklists for examples <a href="http://gigadb.org/site/guide" target="_blank">http://gigadb.org/site/guide</a>YesAdditional CommentsAll that was provided was consistent.Is the data acquisition clear, complete and methodologically sound?NoAdditional CommentsSome clarification is needed (was the same sample used for the genome and transcriptome assembly? Were the different tissues processed in the same way? What software were used for all the bioinformatics steps? What were all the parameters and filters used for genome and transcriptome assembly and annotation?) I left specific suggestions in a file with additional comments to the authors.Is there sufficient detail in the methods and data-processing steps to allow reproduction?NoAdditional CommentsSoftware versions, citations, and parameters are missing from the methods section. Some results refer to methods not explained in the methods section. Is there sufficient data validation and statistical analyses of data quality? YesAdditional CommentsIs the validation suitable for this type of data?YesAdditional CommentsMore details on the BlobTools parameters used are needed. Is there sufficient information for others to reuse this dataset or integrate it with other data?NoAdditional CommentsSupplementary tables were mentioned but not provided (at least not for review). There is enough information for others to reuse the genome data, although more information in the methods section (as mentioned above) and a metadata file would make this even more useful. There is no mention of where the transcriptome has been deposited, and an extremely brief mention to how it was assembled (e.g., no details on parameters used or software versions). Any Additional Overall Comments to the AuthorPlease include all citations in the reference list.RecommendationMinor Revision